# Contrasting Toxin Selectivity between the Marine Pufferfish *Takifugu pardalis* and the Freshwater Pufferfish *Pao suvattii*

**DOI:** 10.3390/toxins11080470

**Published:** 2019-08-10

**Authors:** Wei Gao, Yoko Kanahara, Misako Yamada, Ryohei Tatsuno, Hiroyuki Yoshikawa, Hiroyuki Doi, Tomohiro Takatani, Osamu Arakawa

**Affiliations:** 1Graduate School of Fisheries and Environmental Sciences, Nagasaki University. 1-14, Bunkyo-machi, Nagasaki, Nagasaki 852-8521, Japan; 2National Fisheries University, Japan Fisheries Research and Education Agency. 2-7-1, Nagatahonmachi, Shimonoseki, Yamaguchi 759-6595, Japan; 3Nifrel, Osaka Aquarium Kaiyukan. 2-1, Senribanpakukoen, Suita, Osaka 565-0826, Japan

**Keywords:** tetrodotoxin (TTX), paralytic shellfish poison (PSP), saxitoxin (STX), pufferfish, *Takifugu pardalis*, *Pao suvattii*

## Abstract

To clarify the differences in toxin selectivity between marine and freshwater pufferfish, we conducted experiments in artificially reared nontoxic specimens of *Takifugu pardalis* (marine) and *Pao suvattii* (freshwater) using tetrodotoxin (TTX) and paralytic shellfish poison (PSP; decarbamoylsaxitoxin (dcSTX) or saxitoxin (STX)). *T. pardalis* specimens were administered feed homogenate containing TTX or dcSTX (dose of toxin, 55.2 nmol/fish) and *P. suvattii* specimens were administered feed homogenate containing TTX + STX (dose of each toxin, 19.2 nmol/fish) by oral gavage. The toxin content in the intestine, muscle, skin, liver, and gonads was quantified after 24 and 48 or 72 h. In *T. pardalis*, TTX administered into the intestine was absorbed into the body and transferred and retained mainly in the skin and liver, while dcSTX was hardly retained in the body, although it partly remained in the intestine. In strong contrast, in *P. suvattii*, little TTX remained in the body, whereas STX was absorbed into the body and was transferred and retained in the ovary and skin. The findings revealed that TTX/PSP selectivity differs between the marine species *T. pardalis* and the freshwater species *P. suvattii*. *T. pardalis*, which naturally harbors TTX, selectively accumulates TTX, and *P. suvattii*, which naturally harbors PSP, selectively accumulates PSP.

## 1. Introduction

Marine pufferfish of the genus *Takifugu*, including *Takifugu pardalis*, contain the potent neurotoxin tetrodotoxin (TTX) as a main toxic component [1]. The liver and ovary are usually strongly toxic [2], and food poisoning due to the consumption of these fish often occurs in Japan [3]. TTX is distributed not only in pufferfish but also in various other organisms, many of which are toxified exogenously by the food chain starting with TTX-producing bacteria [1]. Therefore, pufferfish that are artificially reared with nontoxic feed from hatching do not contain TTX [4]. Using such nontoxic cultured fish, we conducted various TTX administration experiments [5,6,7,8,9,10] and found that the internal TTX kinetics in pufferfish are unique from the intestine to the liver and from the liver to the skin or ovary. The kinetics may also differ between males and females and change with the growth and/or maturation of individual fish [7,9].

On the other hand, small-sized freshwater pufferfish, including *Pao suvattii* (formerly known as *Tetraodon suvattii*), contain paralytic shellfish poison (PSP) as the major toxic component in their skin and ovary, and consumption of these fish occasionally causes food poisoning in Southeast Asian countries such as Thailand and Cambodia [11,12,13,14]. PSP is a group of neurotoxins produced by toxic dinoflagellates of the genera *Alexandrium*, *Gymnodinium*, and *Pyrodinium* in marine environments and by toxic cyanobacteria of the genera *Anabaena*, *Cylindrospermopsis*, *Aphanizomenon*, *Planktothrix*, and *Lyngbya* in freshwater environments [15]. The main component of PSP, saxitoxin (STX), has an almost equivalent molecular size and mechanism of action to TTX [15]. *Sphoeroides* pufferfish in Florida [16] and the marine *Arothron* pufferfish in the Philippines [17] and Japanese coastal waters [18] are also highly toxified by PSP. In addition, small amounts of PSP are detected in edible marine pufferfish of the genus *Takifugu* [19,20,21], raising a food hygiene issue in Japan. Marine pufferfish are thought to accumulate PSP via the food chain starting from PSP-producing dinoflagellates [16,17], while freshwater pufferfish are thought to accumulate PSP from PSP-producing cyanobacteria [13]. It is unclear, however, whether the marine *Takifugu* pufferfish can accumulate PSP equally to freshwater pufferfish or whether freshwater pufferfish can accumulate TTX equally to marine *Takifugu* pufferfish if the toxin supply in the food is sufficient. In this study, we administered TTX and/or PSP to artificially reared nontoxic specimens of *T. pardalis* and *P. suvattii* to clarify the differences in toxin selectivity between marine and freshwater pufferfish.

## 2. Results

The toxin concentration in the examined organs of *T. pardalis* and *P. suvattii* at 24 and 48 or 72 h after TTX and/or PSP administration is shown in Figure 1 and Table A1 and Table A2. In *T. pardalis*, TTX administered into the intestine by oral gavage was taken up into the body and transferred mainly to the skin, ovary, and liver. The TTX concentration was highest in the skin at 24 h and in the ovary at 72 h, and the TTX concentration in the liver was generally lower than that in the ovary and skin. At 24 h, a low concentration of TTX remained in the intestine, and at 72 h, the concentration in the intestine further decreased to a trace level. On the other hand, the dcSTX concentration in the intestine remained high even at 72 h and was detected at low concentrations only in the gonads among the other organs examined. In *P. suvattii,* in strong contrast to *T. pardalis*, no TTX was detected in any organs other than the intestine at 24 and 48 h, while STX was taken up into the body and transferred only to the ovary and skin. Interestingly, a much higher concentration of STX than TTX remained in the intestine, similar to *T. pardalis*.

The relative toxin amount (% of the administered toxin) in each organ of *T. pardalis* and *P. suvattii* is shown in Figure 2 and Table A3 and Table A4. In *T. pardalis*, 16.6–55.1% (see “Total” in Table A3) of the administered TTX was absorbed and retained in the body at 24 and 72 h, with most (82.7–95.1% of the “Total”) distributed in the skin and the rest distributed mainly in the liver. At 72 h, little toxin (~0.1% of the administered toxin) remained in the intestine. On the other hand, dcSTX was not retained in the rest of the body except for in the gonads, in which little toxin accumulated (0.2% in females and 0.01% in males), and 8.4–10.3% of the dose remained in the intestine. In strong contrast, in *P. suvattii*, little TTX (0.03–0.5%) remained in the body at 24 and 48 h, whereas STX (29.3–35.0% in females and 2.7–12.6% in males) was absorbed by the body and retained in the ovary and skin. The intestine contained 0.6–6.8% of the administered dose.

## 3. Discussion

Our findings strongly indicate that the marine species *T. pardalis* and the freshwater species *P. suvattii* are endowed with contrasting TTX/PSP selectivity. Due to the scarcity of PSP preparations and of artificially reared pufferfish specimens, or as a matter of practical convenience, the two toxin administration experiments conducted in the present study differed with respect to the experimental conditions, such as the PSP components, dose, administration method (TTX and PSP administered alone or in combination), number of test fish, and rearing period, thereby preventing a quantitative comparison of the selectivity between species. Nevertheless, our findings clearly demonstrate for the first time that *T. pardalis*, which naturally harbors TTX, selectively accumulates TTX, and *P. suvattii*, which naturally harbors PSP, selectively accumulates PSP.

In *T. pardalis*, little dcSTX, and in *P. suvattii*, little TTX was retained in any organs other than the intestine. There are three possible reasons for this: (1) the intestine serves as a barrier allowing little toxin to be absorbed into the body; (2) the toxin is absorbed into the body from the intestine but is quickly decomposed and/or excreted without being taken up into the liver and skin; and (3) the toxin is absorbed into the body from the intestine and taken up into certain organs but is converted to other analogues that were not detectable in the present study. Nagashima et al. [22,23] and Matsumoto et al. [24], using a tissue slice culture method, demonstrated that the liver tissues of marine pufferfish of the genera *Takifugu* and *Lagocephalus* take up considerable amounts of TTX, unlike general marine fish, but do not take up PSP like general marine fish. This finding seems to support the second possibility but does not rule out the first possibility. To clarify this point, we are currently investigating whether the TTX/PSP uptake ability of the intestine and/or skin can be evaluated using the tissue slice culture method. This may also help to clarify, to some extent, why PSP is more likely than TTX to remain in the intestine in both *T. pardalis* and *P. suvattii*.

TTX and its analogues 4-*epi*TTX and 4,9-anhydroTTX are converted to each other and exist in equilibrium in an approximate ratio of 8:1:1 in an aqueous acidic solution [25]. Under the analytical conditions of the present study, 4-*epi*TTX was detectable with TTX but was below the limit of quantification (LOQ) in any of the organs examined. Yasumoto et al. and Yotsu-Yamashita et al. [26,27,28,29,30] separated many TTX analogues from various pufferfish, and they estimated the biosynthetic and metabolic pathways of TTX through a series of oxidations as 5,6,11-trideoxyTTX → 5,11-dideoxyTTX or 6,11-dideoxyTTX → 5-deoxyTTX, 11-deoxyTTX or 6-deoxyTTX → TTX → 11-*nor*TTX-6-ol → 11-oxoTTX. It is possible that TTX administered to the pufferfish is oxidized to 11-*nor*TTX-6-ol or 11-oxoTTX. These analogues, as well as 4,9-anhydroTTX, are all minor components coexisting with TTX in pufferfish [21,28,29,31,32], and it is unlikely that all TTX taken up into the body is converted to these analogues. On the other hand, conversions of PSP components (hydrolysis of *N*-sulfocarbamoyl toxins and reductive elimination of the sulfate group at C-11 or hydroxyl at N-1) occur in PSP-contaminated bivalves or toxic xanthid crabs, but in this case, dcSTX and STX are rather stable final products [33,34,35,36]. In the present study, we could not exclude the possibility that TTX or dcSTX/STX are converted to nondetectable metabolites as the analysis target was limited to the administered toxins. It is unlikely, however, that TTX or dcSTX/STX, which are the main components of the toxins naturally harbored in pufferfish, are completely converted to other components in such a short period of time. Further studies are needed to evaluate the possibility that the TTX or STX analogues were not detectable.

Ngy et al. [14] intramuscularly administered TTX and PSP into artificially reared specimens of the freshwater pufferfish *Leiodon* (formerly known as *Tetraodon*) *tergidus* and found that PSP rapidly transferred from the muscle to the skin, where it accumulated. On the other hand, all the fish administered TTX died within 3–4 h, and approximately half the TTX dose remained in the muscles of the dead fish. Although the toxin administration method differed from that in the present study, *L. tergidus* is considered to have similar toxin selectivity as *P. suvattii*. In addition, this finding indicates that freshwater pufferfish are not very resistant to TTX. Marine pufferfish have much stronger resistance to TTX than do general fish, but their resistance to PSP is not very strong [37,38,39]. Thus, toxin selectivity could be closely related to toxin resistance.

Our previous TTX administration experiments using marine pufferfish revealed that TTX administered into the pufferfish body is transferred first to the liver and then to the skin via the bloodstream. Tatsuno et al. [9] conducted a TTX administration experiment using specimens of the marine pufferfish *Takifugu rubripes* of different ages. They reported that at an early age, *T. rubripes* transport TTX mainly to their skin to accumulate or eliminate because the liver is undeveloped and has low TTX-accumulating ability, but as they grow, the liver develops and TTX is eventually stored in the liver. In the present study, TTX absorbed by *T. pardalis* was transferred mainly to the liver, skin, and ovary, but in terms of the relative amount of toxin, most of the toxin retained in the body was distributed to the skin. In *T. pardalis*, toxicity of the liver and ovary is generally highest, followed by the skin [1]. The artificially reared *T. pardalis* specimens used in the present study may have had an insufficiently developed liver.

Wang et al. [7] reported that intramuscular administration of TTX to hybrid specimens produced by crossbreeding *T. rubripes* with *Takifugu alboplumbeus* (formerly known as *Takifugu niphobles*), which matures earlier than *T. rubripes*, is first taken up in the liver and then transferred to and accumulated in the skin in male fish and the ovary in female fish. This finding suggests that TTX kinetics in the pufferfish body are strongly influenced by sexual maturation. Gao et al. [40] reported that in wild *T. pardalis* specimens, the TTX concentration in the ovary increases from the yolk vesicle stage to the yolk globule stage, and the TTX amount in the ovary increases during the yolk globule stage. The TTX distribution inside the body of *T. pardalis* specimens at 72 h after TTX administration in the present study was similar to that of immature wild specimens in the yolk vesicle stage. Although the TTX concentration was highest in the ovary, the relative TTX amount (toxin amount per individual) was highest in the skin, and that in the ovary was very low. The female specimens used in this study were considered to be immature, with a gonadosomatic index (GSI) of 0.29 ± 0.06, which is comparable to that of wild specimens at the yolk globule stage (~0.3) [40].

Wild specimens of *P. suvattii* have highly toxic skin and ovaries [11]. In the present study, STX absorbed into the body was mainly transferred to the skin and ovary like in wild specimens. One potential reason for the higher relative STX amount in the *P. suvattii* ovary than the relative TTX amount in the *T. pardalis* ovary is that the *P. suvattii* female specimens, whose GSI (9.6 ± 2.1) was much higher than that of the *T. pardalis* specimens, were more mature than the *T. pardalis* specimens, despite being the same age.

Yotsu-Yamashita et al. [41] separated pufferfish STX and TTX binding protein (PSTBP) from the blood plasma of *T. pardalis*. They investigated the localization of PSTBP in the tissues of *T. pardalis* using an immunohistochemical technique and concluded that PSTBP functions as a toxin transporter in the blood and is involved in toxin absorption in the intestine, as well as toxin accumulation in the liver, ovary, and skin [42]. Tatsuno et al. [43] reported four isoform genes homologous to PSTBP in *T. rubripes*, two of which are expressed in the liver. Such PSTBP homologous protein isoforms are also found in other toxic species of *Takifugu* but not in nontoxic species and general fish [44,45]. The presence or absence of these protein isoforms in freshwater pufferfish, however, is unknown. It would be interesting to clarify whether freshwater pufferfish possess similar isoforms and the role the PSTBP homologous isoforms play in the contrasting TTX/PSP selectivity between marine and freshwater pufferfish. Studies along this line are in progress.

## 4. Materials and Methods

### 4.1. Pufferfish Specimens

Artificially reared nontoxic 12-month-old *T. pardalis* (body length, 10.2 ± 0.5 cm; body weight, 36.7 ± 6.2 g; *n* = 30 (11 females and 19 males)) and *P. suvattii* (body length, 8.3 ± 0.4 cm; body weight, 28.0 ± 3.1 g; *n* = 8 (5 females and 3 males)) were used for the toxin administration experiments, as described below. The GSI (100 × gonad weight/body weight) in *T. pardalis* was 0.29 ± 0.06 in females and 0.35 ± 0.16 in males, and that in *P. suvattii* was 9.6 ± 2.1 in females and 0.45 ± 0.16 in males.

### 4.2. Preparation of Toxic Feed Homogenate

TTX extracted from the ovaries of wild *T. rubripes* and PSP extracted from the xanthid crab *Zosimus aeneus* were purified by solvent partitioning, activated charcoal treatment, and Bio-Gel P-2 (Bio-Rad Laboratories, Inc., Hercules, CA, USA) and Bio-Rex 70 (Bio-Rad Laboratories, Inc., Hercules, CA, USA) column chromatography according to the previously reported method [46,47], affording TTX (purity >60%), and dcSTX and STX (purity >80%) as essentially single components (content of 4-*epi*TTX in the TTX preparation, <2%). PSP detected in marine and freshwater pufferfish consists of STX as the main component and dcSTX as a secondary component. In our experiments, we administered dcSTX to *T. pardalis* and STX to *P. suvattii* because the availability of STX was insufficient.

In previous TTX administration experiments using marine pufferfish [6,7,8], good results were obtained at doses of 1.3–2.6 nmol/g body weight. On this basis, we administered the toxin at a dose of ~1.5 nmol/g body weight (~55 nmol/fish) in the present study. In *P. suvattii*, we administered TTX and STX as a mixture because only eight fish were available. Therefore, we set the dose of TTX + STX to ~1.4 nmol/g body weight (each toxin ~20 nmol/fish), considering the fact that in the natural environment, *P. suvattii* is generally less toxic than *T. pardalis* [1,11], and that a dose of 2.3 nmol/g body weight TTX kills the freshwater pufferfish *L. tergidus* [14].

An aqueous solution of TTX or dcSTX was mixed with artificial feed for marine juvenile fish (Otohime C2, Marubeni Nisshin Feed Co., Ltd., Tokyo, Japan) at a ratio of 2/1 (v/w) and homogenized, and then feed homogenates containing either TTX or dcSTX (552 nmol/mL) were administered to *T. pardalis*. For *P. suvattii*, the feed homogenate was similarly prepared to contain a mixture of TTX and STX (192 nmol/mL each).

### 4.3. Toxin Administration to *T. pardalis*

*T. pardalis* specimens were divided into three groups of 10 individuals—two TTX administration groups and one dcSTX administration group—and then maintained separately in three aerated 90 L tanks with artificial seawater at 25 °C. Each specimen was administered TTX- or dcSTX-containing feed homogenate (dose of toxin, 55.2 nmol/fish) by oral gavage [8] and immediately returned to the tank. In previous TTX administration experiments using marine pufferfish [6,7,8], we observed that the administered TTX was first transferred to the liver and then to the skin and ovary, and the toxin distribution inside the body stabilized within 72 h after toxin administration. Therefore, one TTX administration group and the dcSTX administration group were collected at 72 h, and toxin quantification was performed as described below to compare the toxin distribution in the body. As the amount of TTX available was sufficient, a 24 h TTX administration group was also used to confirm the transition of the toxin distribution inside the body.

### 4.4. Toxin Administration to *P. suvattii*

*P. suvattii* specimens were maintained in an aerated 60 L tank with dechlorinated tap water at 25 °C. Each specimen was administered the feed homogenate containing TTX and STX (dose of each toxin, 19.2 nmol/fish) by oral gavage and immediately returned to the tank. In a previous study in which PSP was administered to *L. turgidus* [14], the amount of toxin in organs other than the skin rapidly decreased 12 h after toxin administration and remained in only trace amounts at 48 h after administration. We did not have enough *P. suvattii* to collect data on the toxin distribution at 72 h. Therefore, four specimens were randomly collected at 24 or 48 h after toxin administration, and toxin quantification was performed as described below to compare the distribution of each toxin.

### 4.5. Toxin Quantification

In each fish, the intestine, muscle, skin, liver, and gonads (testes/ovaries) were removed and extracted with 0.1 M HCl [14], passed through an HLC-DISK membrane filter (0.45 µm, Kanto Chemical Co., Inc., Tokyo, Japan), and submitted to liquid chromatography tandem mass spectrometry (LC-MS/MS) for TTX [40] and/or high-performance liquid chromatography with post-column fluorescence derivatization (HPLC-FLD) for PSP [18].

In the LC-MS/MS analysis, chromatography was carried out using an Alliance 2690 Separations Module (Waters, Milford, MA, USA) with a Mightysil RP-18 GP column (2.0 × 250 mm, particle size 5 µm, Kanto Chemical Co., Inc., Tokyo, Japan) and mobile phase comprising 30 mM heptafluorobutyric acid in 1 mM ammonium acetate buffer (pH 5.0) at a flow rate of 0.2 mL/min. The eluate was introduced into a Quattro microTM API detector (Waters, Milford, MA, USA) in which the TTX was ionized by positive-mode electrospray ionization with a desolvation temperature of 350 °C, source block temperature of 120 °C, and cone voltage of 50 V and monitored at *m*/*z* 162 (for quantitative) and 302 (for qualitative) as product ions (collision voltage 38 V) with *m*/*z* 320 as a precursor ion through a MassLynxTM NT operating system (Waters, Milford, MA, USA). The limit of detection and LOQ of TTX were 0.0009 nmol/mL (0.003 nmol/g tissue; *S*/*N* = 3) and 0.003 nmol/mL (0.009 nmol/g tissue; *S*/*N* = 10), respectively.

HPLC-FLD was performed using an Alliance 2690 Separations Module with a LiChroCART Superspher RP18(e) column (4.6 × 250 mm, particle size 4 µm, Merck, Darmstadt, Germany) and a mobile phase comprising 2 mM heptanesulfonic acid in 4% acetonitrile–30 mM ammonium phosphate buffer (pH 7.3) at a flow rate of 0.8 mL/min. The eluate was continuously mixed with 50 mM periodic acid and 0.2 M KOH containing 1 M ammonium formate and 50% formamide and heated at 65 °C. The formation of fluorophores was monitored at 392 nm with 336 nm excitation. The limit of detection of both dcSTX and STX was 0.007 nmol/mL (0.02 nmol/g tissue; *S*/*N* = 3) and the LOQ of both dcSTX and STX was 0.02 nmol/mL (0.06 nmol/g tissue; *S*/*N* = 10).

## Figures and Tables

**Figure 1 toxins-11-00470-f001:**
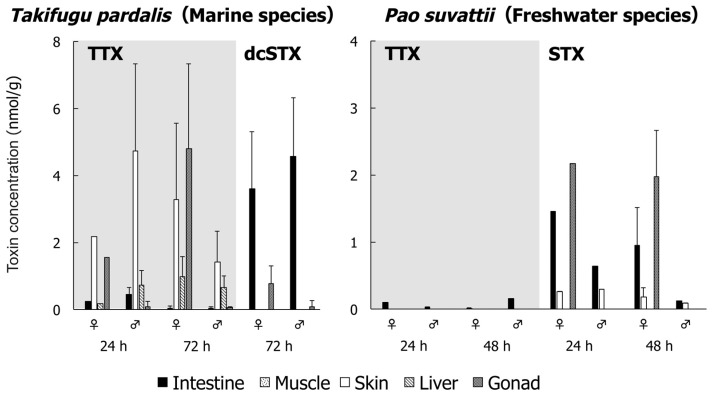
Toxin concentration in each organ of *Takifugu pardalis* (left) and *Pao suvattii* (right) at 24 and 48 or 72 h after tetrodotoxin (TTX) and/or paralytic shellfish poison (PSP; decarbamoylsaxitoxin (dcSTX) or saxitoxin (STX)) administration. Data are shown as means (columns) and SDs (error bars).

**Figure 2 toxins-11-00470-f002:**
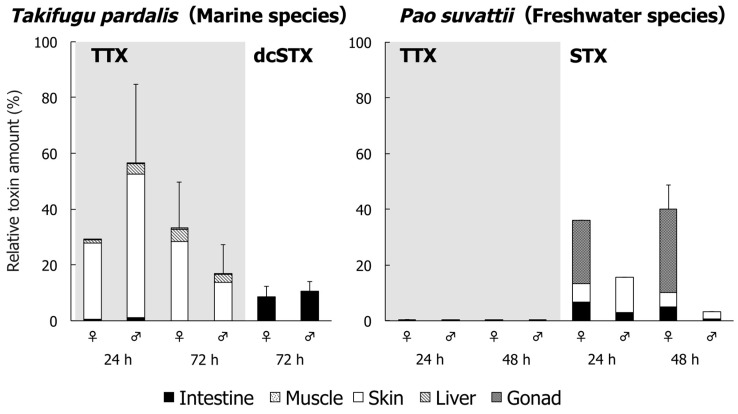
Relative toxin amount (% of the administered toxin) in each part of *T. pardalis* (left) and *P. suvattii* (right) at 24 and 48 or 72 h after TTX and/or PSP administration. Data are shown as means (columns) and SDs (error bars).

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
