# Peer review of "Contrasting Toxin Selectivity between the Marine Pufferfish Takifugu pardalis and the Freshwater Pufferfish Pao suvattii"

_toxins, 2019, doi:10.3390/toxins11080470_

Round 1

Reviewer 1 Report

Please specify the dimensions of columns particle size for both LC-MS/MS and HPLC-FLD methods

Author Response

Thank you very much for your high evaluation of our manuscript. We revised our manuscript according to your comment (revised parts are indicated in blue font).

Please specify the dimensions of columns particle size for both LC-MS/MS and HPLC-FLD methods

The columns particle size is now specified in the revised manuscript (P6L240, 250).

Reviewer 2 Report

The manuscript constitutes an interesting report, as new data on pufferfish toxicity and especially on their selectivity trends regarding toxin accumulation are always of interest to marine toxin research. A rather substantial revision is however required to address the comments indicated below, in order to make the manuscript more scientifically sound and reader friendly. A slight improvement in English is also recommended.

Points for revision:

Major issues to be answered (anywhere in the manuscript convenient for the authors):

1) Why were the specific doses and testing times selected, taken into account the almost 3-fold difference between the different pufferfish species tested? The authors mention the issue of material availability but still this is not a good enough reason, as equally divided lower doses could have been selected to confront this problem. Is this related to the usual toxin amounts found in the specific species or in their usual natural environment? Is there a different accumulation trend reported regarding marine or freshwater species? This should be justified by the authors in order to verify that the experimental design is correct.

2) Why in particular were two different PSP analogues (STX, dcSTX) selected to be tested on the two different pufferfish species? Is there a previously reported PSP analogue selectivity trend which guided this selection for the experiment? This should also be justified by the authors.

3) In the case of TTX: it is certain that the toxic material included in the feed for ingestion (feed homogenate) contained a variety of different TTX analogues, as it was derived from a natural source (pufferfish ovaries). It is also evident that only the parent TTX was determined in the chemical analysis, as only the transition of 320/162 is mentioned in the LC-MS/MS analysis. How can the authors exclude the possibility that the toxin ingested is metabolized to different, non-determinable, analogues, which could the main reason that is not anymore detectable after some time? This could only be answered if all known (or possibly contained in the homogenate) analogues were included in the LC-MS/MS analysis. This issue is also strongly related to distribution of the toxin among the different tissues and should definitely be taken into account when discussing these findings.

4) The same problem, as above, should be answered in the case of PSP toxins, taking into account the toxin transformations possible after ingestion (or injection) of the toxin into the pufferfish organism and the different analogues/metabolites derived.

Other specific comments:

1. Introduction:

- Page 1, line 41: within the concept of this manuscript, is only Alexandrium spp. responsible for PSP toxicity? Is this the case also in freshwater environments, since a freshwater pufferfish species, Pao suvattii, is concerned?

2. Results:

- It would be better to provide all the results in tables, rather than diagrams, as it is easier to compare data and observe the differences. It is quite hard to follow the comparative proportions with the diagrammatic presentation.

- It is necessary to provide the method quality data for both methods used (LC-MS/MS for TTX and HPLC-FLD for PSPs). Specifically, the methods’ LOD, LOQ and recovery should definitely be provided, as the readers should be able to realize what exactly the “absence” or low levels of toxins mean (e.g. absence of TTX in P. suvattii after administration, low levels of dcSTX in T. pardalis).

- What was the % proportion of TTX in the gonads of T. pardalis? Was there a difference with regard to distribution between male and female individuals? Please provide the numbers in tables in all cases to indicate the differences.

3. Discussion:

- Please discuss the tissue distribution in terms of the possible metabolism of toxins, as indicated above. Also, please provide details, if reported in the literature, in comparison to the naturally contaminated specimens, e.g. is the same trend of PSP toxin accumulation in ovary and skin observed in naturally contamimated P. suvattii specimens? Is the same trend of TTX accumulation in skin, ovary and liver observed in naturally contaminated specimens of T. pardalis?

4. Materials and Methods

- Page 4, lines 139-142: The sexual maturity status of the experimental pufferfish should be mentioned, as the existence of gonads is mentioned.

- Page 4, lines 143-150: Please justify, as indicated above, the reasons for the different doses and different analogues administered to the two different pufferfish species.

- Pages 4-5, lines 151-163: Please justify, as indicated above, the basis for the different sampling times selected in each case (e.g. dcSTX only at 72h).

- Please provide details on animal experimentation licensing, these are required in order to confront the ethical issues related to animal experimentation.

- Page 5, lines 176-178: please clarify if only one transition was monitored (320/162), as a LC-MS/MS method is indicated, where at least two transitions should be monitored. The selected transition is also related to the determination of 4-epiTTX which should be mentioned in the text. Why were the other TTX analogues not included in the study?

Author Response

Many thanks for your valuable comments. We revised our manuscript according to the comments, as indicated below (revised parts are indicated in blue font). The revised manuscript was checked by a professional English editing service (edited parts are not shown).

The manuscript constitutes an interesting report, as new data on pufferfish toxicity and especially on their selectivity trends regarding toxin accumulation are always of interest to marine toxin research. A rather substantial revision is however required to address the comments indicated below, in order to make the manuscript more scientifically sound and reader friendly. A slight improvement in English is also recommended.

 Points for revision:

Major issues to be answered (anywhere in the manuscript convenient for the authors):

1) Why were the specific doses and testing times selected, taken into account the almost 3-fold difference between the different pufferfish species tested? The authors mention the issue of material availability but still this is not a good enough reason, as equally divided lower doses could have been selected to confront this problem. Is this related to the usual toxin amounts found in the specific species or in their usual natural environment? Is there a different accumulation trend reported regarding marine or freshwater species? This should be justified by the authors in order to verify that the experimental design is correct.

As now described in the ‘Materials and methods’ (P5L199-205), the specific doses and testing times were selected on the basis of findings from toxin administration experiments we previously conducted. Material availability was also a major factor. As the sex of the pufferfish cannot be distinguished by appearances, it is impossible to adjust the ratio of females/males in each group. Therefore, large groups of T. pardalis were used for the administration experiment.

It is virtually impossible to simultaneously obtain healthy artificially reared marine and freshwater pufferfish, and to conduct an experiment under the same conditions. Therefore, some conditions differed in the two independently conducted toxin administration experiments, and their results cannot be directly compared. Nevertheless, toxin selectivity, which was the main focus of this research, was clearly demonstrated.

2) Why in particular were two different PSP analogues (STX, dcSTX) selected to be tested on the two different pufferfish species? Is there a previously reported PSP analogue selectivity trend which guided this selection for the experiment? This should also be justified by the authors.

PSP detected in marine pufferfish and freshwater pufferfish both contain STX as the main component and dcSTX as a minor component. It would have been preferable to administer STX to T. pardalis, but we had to use dcSTX because we could not prepare a sufficient amount of STX, and this is now explained in the text (P5L196-198). Among the PSP components, the chemical properties and chromatographic behavior of dcSTX are very similar to those of STX. Therefore, we think that dcSTX is an acceptable substitute for STX in these experiments.

3) In the case of TTX: it is certain that the toxic material included in the feed for ingestion (feed homogenate) contained a variety of different TTX analogues, as it was derived from a natural source (pufferfish ovaries).

The TTX, dcSTX, and STX used in the present study were each separated as essentially a single component using an ion exchange column (Bio-Rex 70) (P5L191-196, please see the references [46,47] for details), although the purity was not confirmed by NMR.

It is also evident that only the parent TTX was determined in the chemical analysis, as only the transition of 320/162 is mentioned in the LC-MS/MS analysis. How can the authors exclude the possibility that the toxin ingested is metabolized to different, non-determinable, analogues, which could the main reason that is not anymore detectable after some time? This could only be answered if all known (or possibly contained in the homogenate) analogues were included in the LC-MS/MS analysis. This issue is also strongly related to distribution of the toxin among the different tissues and should definitely be taken into account when discussing these findings.

Due to the lack of standard TTX analogues, only parent TTX was analyzed in the present study. Certainly, we cannot exclude the possibility that the ingested toxin is metabolized to different, non-determinable analogues, which could be the main reason that it is no longer detectable after some time. We now discuss this issue in the ‘Discussion’ (P3L102-104, P4L112-130).

4) The same problem, as above, should be answered in the case of PSP toxins, taking into account the toxin transformations possible after ingestion (or injection) of the toxin into the pufferfish organism and the different analogues/metabolites derived.

This issue is also now addressed in the ‘Discussion’ (P4L122-130).

Other specific comments:

Introduction:

- Page 1, line 41: within the concept of this manuscript, is only Alexandrium spp. responsible for PSP toxicity? Is this the case also in freshwater environments, since a freshwater pufferfish species, Pao suvattii, is concerned?

According to this comment, we listed the genus names for the dinoflagellates that produce PSP in marine environments and the cyanobacteria that produce PSP in freshwater environments (P1L42-P2L44).

Results:

- It would be better to provide all the results in tables, rather than diagrams, as it is easier to compare data and observe the differences. It is quite hard to follow the comparative proportions with the diagrammatic presentation.

We believe that the diagrams are helpful to comprehensively demonstrate the differences in the toxin selectivity. Tables were added to the Appendix (P7, 8) to compare the detailed data, and the numbers in the text were slightly revised accordingly (P2L75-P3L83).

- It is necessary to provide the method quality data for both methods used (LC-MS/MS for TTX and HPLC-FLD for PSPs). Specifically, the methods’ LOD, LOQ and recovery should definitely be provided, as the readers should be able to realize what exactly the “absence” or low levels of toxins mean (e.g. absence of TTX in P. suvattii after administration, low levels of dcSTX in T. pardalis).

The limit of detection and limit of quantification are now provided (P6L246-248, 254-256). We currently have no data on recovery. Both LC-MS/MS for TTX and HPLC-FLD for PSP are generally and widely used reliable methods, however, and we consider that the problem is minimal with regard to the aim of the study.

- What was the % proportion of TTX in the gonads of T. pardalis? Was there a difference with regard to distribution between male and female individuals? Please provide the numbers in tables in all cases to indicate the differences.

The % proportion of TTX in the gonads differed between the females (0.2%-0.7%) and males (0.01%). The numbers are now provided both in the revised text (P3L79-80) and in Table A3 (P8).

Discussion:

- Please discuss the tissue distribution in terms of the possible metabolism of toxins, as indicated above. Also, please provide details, if reported in the literature, in comparison to the naturally contaminated specimens, e.g. is the same trend of PSP toxin accumulation in ovary and skin observed in naturally contamimated P. suvattii specimens? Is the same trend of TTX accumulation in skin, ovary and liver observed in naturally contaminated specimens of T. pardalis?

We now discuss the tissue distribution in the revised manuscript, as suggested (P4L140-P5L170).

Materials and Methods

- Page 4, lines 139-142: The sexual maturity status of the experimental pufferfish should be mentioned, as the existence of gonads is mentioned.

The data for GSI, which is an indicator of maturity, were added to the revised manuscript (P5L188-189).

- Page 4, lines 143-150: Please justify, as indicated above, the reasons for the different doses and different analogues administered to the two different pufferfish species.

The reasons for the use of different doses and different analogues are now explained in the revised manuscript (P5L196-205).

- Pages 4-5, lines 151-163: Please justify, as indicated above, the basis for the different sampling times selected in each case (e.g. dcSTX only at 72h).

The basis for the different sampling times is now explained in the revised manuscript (P6L215-222, 226-231).

- Please provide details on animal experimentation licensing, these are required in order to confront the ethical issues related to animal experimentation.

We essentially conduct animal experiments following the Guidelines for Animal Experimentation of Nagasaki University and the Regulations of the Animal Care and Use Committee, Nagasaki University. These guidelines and regulations, however, currently do not include fish, amphibians, and invertebrates.

- Page 5, lines 176-178: please clarify if only one transition was monitored (320/162), as a LC-MS/MS method is indicated, where at least two transitions should be monitored. The selected transition is also related to the determination of 4-epiTTX which should be mentioned in the text. Why were the other TTX analogues not included in the study?

We also monitored 320/302 qualitatively, which is mentioned in the text (P6L244-245). While 4-epiTTX was detectable, it was below the limit of quantification in all organs examined (P4L113-115). The aim of the study was not to examine how TTX is metabolized when administered to pufferfish, but to clarify if there is a difference in the selectivity for TTX and PSP. Therefore, we simply traced the administered TTX itself. Furthermore, precise analysis is difficult because we do not have standard TTX analogues. Although comprehensive analysis of TTX analogues may provide more information, we believe that the aim of the study could be sufficiently achieved by analysis of only TTX.

Round 2

Reviewer 2 Report

I am generally very satisfied with the revision and the point-by-point explanations provided by the authors.

I only suggest one more minor correction in Page 3, line 104, where the text "converted to other analogues that are not detectable" should be changed to "converted to other analogues that were not detectable in the present study". This is because it is possible to detect several of these analogues by including their relevant transitions in the LC-MS/MS method for TTXs (even when standards not exist), or some of them in the HPLC-FLD method for PSPs (at least those for which standards do exist).

Author Response

Thank you very much for your comments. We revised our manuscript according to the comments (P3L104, red font).